# *Salmonella enterica*’s “Choice”: Itaconic Acid Degradation or Bacteriocin Immunity Genes

**DOI:** 10.3390/genes11070797

**Published:** 2020-07-15

**Authors:** Rolf D. Joerger

**Affiliations:** Department of Animal and Food Sciences, University of Delaware, Newark, DE 19716, USA; rjoerger@udel.edu

**Keywords:** *Salmonella*, itaconic acid degradation, bacteriocin immunity genes

## Abstract

Itaconic acid is an immunoregulatory metabolite produced by macrophages in response to pathogen invasion. It also exhibits antibacterial activity because it is an uncompetitive inhibitor of isocitrate lyase, whose activity is required for the glyoxylate shunt to be operational. Some bacteria, such as *Yersinia pestis,* encode enzymes that can degrade itaconic acid and therefore eliminate this metabolic inhibitor. Studies, primarily with *Salmonella enterica* subspecies enterica serovar Typhimurium, have demonstrated the presence of similar genes in this pathogen and the importance of these genes for the persistence of the pathogen in murine hosts. This minireview demonstrates that, based on Blast searches of 1063 complete *Salmonella* genome sequences, not all Salmonella serovars possess these genes. It is also shown that the growth of *Salmonella* isolates that do not possess these genes is sensitive to the acid under glucose-limiting conditions. Interestingly, most of the serovars without the three genes, including serovar Typhi, harbor DNA at the corresponding genomic location that encodes two open reading frames that are similar to bacteriocin immunity genes. It is hypothesized that these genes could be important for *Salmonella* that finds itself in strong competition with other *Enterobacteriacea* in the intestinal tract—for example, during inflammation.

## 1. *Salmonella* and Itaconic Acid

Itaconic acid and bacteriocins are among the many compounds that *Salmonella* species must contend with in their environments and hosts. Itaconic acid (2-methylenesuccinic acid) is a dicarboxylic acid that *Salmonella* may, on occasion, encounter in the environment when co-existing with fungal producers of the compound, such as *Aspergillus itaconicus*, *A. terreus*, *Ustilago zeae*, *U. maydis*, or certain *Candida* species (reviewed in [1]). The pathogen might even be exposed to the acid when it finds itself inside marine bivalves such as mussels [2,3] or oysters [4]. More likely, though, *Salmonella* species will encounter the acid when near or inside activated macrophages, since these cells produce and secrete itaconic acid upon stimulation by γ-interferon and lipopolysaccharide [5]. Itaconic acid in activated macrophages is produced by IRG1, the product of immune-responsive gene 1, as was shown through metabolomics and genetic analyses [6].

Cellular concentrations of the acid within lipopolysaccharide (LPS)-treated RAW264.7 mouse macrophages were estimated to be about 8 mM and 3 mM in BV2 retroviral-immortalized microglia cells [6]. At these concentrations, one can expect some minor acidifying effects but more so an antimicrobial effect. First shown with *Pseudomonas indigofera* [7,8,9,10], itaconic acid acts as an inhibitor of isocitrate lyase and inhibits the growth of this organism under glucose-deprived conditions when the glyoxylate shunt is operational [10]. The acid can also inhibit propionyl-CoA carboxylase [11]; therefore, it was proposed that itaconic acid interferes with acetate assimilation in general [12], particularly when bacteria are forced to use fatty acids or cholesterol as carbon sources. That the glyoxylate shunt is also operational in *Salmonella enterica* subspecies enterica serovar Typhimurium (*S. e.* Typhimurium) was shown by mutational analysis in 1987 [13]. While it was not until 2013 that it was demonstrated that an *S. e.* isolate (serovar not specified) could be inhibited by itaconic acid at concentrations of ≥10 mM when grown in medium containing acetate as the sole carbon source [6], earlier studies had already addressed the question of how *Salmonella* deals with itaconic acid. Martin et al. [14] utilized ^14^C-labeled itaconic acid in uptake experiments and experiments using cell-free extracts. The authors also tested whether itaconic acid could be used by *Salmonella* as the sole carbon source. It was discovered that none of the 15 *Salmonella* strains tested were able to grow on the acid as the sole carbon source; however, all but two, designated as *S. typhosa* and *S. pullorum*, showed evidence of uptake of ^14^C itaconic acid. Cell-free extracts of one of the strains, designated *Salmonella chittagong*, showed evidence of the conversion of ^14^C itaconic acid to ^14^C citramalic acid, now known to be an intermediate during itaconic acid degradation. 

In 2005, it was established that a functioning isocitrate lyase, and thus likely the glyoxylate shunt, was required for the persistence of *S. e.* Typhimurium in mice over an eight-week period, but not for acute lethal infection [15]. Therefore, if itaconic acid was also an uncompetitive inhibitor of isocitrate lyase in *Salmonella* strains, as it is in *Pseudomonas aeruginosa* [16], the production of this acid by macrophages could be a means of suppressing the pathogen. Fang et al. [15] speculated that available substrates for *Salmonella* within the phagosome change over the course of the infection. When glucose is present, the full TCA cycle is operational and isocitrate lyase is dispensable; however, when only fatty acids or acetate are available, isocitrate dehydrogenase is phosphorylated, causing its activity to cease, and isocitrate lyase, the enzyme sensitive to itaconic acid, is able to utilize the accumulating isocitrate for the glyoxylate shunt.

## 2. Itaconic Acid Degradation Genes in *Salmonella* and Other Bacteria

To protect their glyoxylate shunt from being inhibited by itaconic acid, some bacteria possess genes encoding enzymes that can degrade the acid. Sasikaran et al. [17] showed that *ripABC* from *Yersinia pestis* encode the three enzymes necessary for itaconic acid degradation. RipA is a succinyl-CoA:itaconate CoA transferase that activates itaconic acid. RipB is an itaconyl-CoA hydratase that hydrates itaconyl-CoA to (*S*)-citramyaly-CoA, and RipC is a (*S*)-citramalyl-CoA lyase that cleaves its substrate to produce acetyl-CoA and pyruvate. The authors found that homologs to *ripABC* are present as a three-gene cluster in several bacteria, including *S. enterica*, *Burkholderia mallei* and *B. speudomallei*, *Mycobacterium uclerans*, *Mycoplasma synoviae* and *Legionalla longbeachae*, whereas in *Pseudomonas aeruginosa* and a number of other bacteria, such as *Brucella* and *Bordetella* spp., the three homologous genes are found within a cluster that contains three additional genes. 

In *S. e.* Typhimurium LT2, stm3117, 3118 and 3119 could encode proteins whose amino acid sequences are 77%, 87% and 72%, respectively, identical to those of their *ripABC*-encoded *Yersinia pestis* counterparts. This level of identity suggests similar functions; however, clear biochemical evidence for the enzymatic function of the *ripABC* homologs in *Salmonella* is not yet available. Nevertheless, the importance of the stm3117 homolog in *S. e.* Enteritidis, then designated *cat2*, for invasion or survival in chicken macrophages was established in 2002 [18]. A few years later, Shah et al. [19] generated signature-tagged mutants of *S. e.* Gallinarum and tested them in competition assays, thereby establishing the stm3118 equivalent as important for virulence of *S. e.* Gallinarum in chickens. The authors also identified the three genes as part of a *Salmonella* pathogenicity island (SPI), designated as SPI-13, located near the gene, *pheV,* encoding tRNA^phe^. The involvement of the three genes in the colonization of RAW 264.7 macrophages by *S. e.* Typhimurium was established in 2006, when Shi et al. [20] conducted a proteomic analysis of the bacteria inside these macrophages and found that the STM3117-3119 proteins were induced upon infection. STM3117 deletion mutants were able to invade macrophages but showed a reduction in replication within the cells. Prior to these studies, Erikson et al. [21] had shown that stm3117, stm3118 and stm3119 transcripts accumulated in *S. e.* Typhimurium inside murine macrophage-like J774-A.1 cells. Hautefort et al. [22] then showed that *S. e.* Typhimurium SL 1344 accumulated stm3117 and stm3119 transcripts when inside macrophages but not when the cells were located within epithelial cells. Conducting competitive infection trials of mice with deletion mutants of *S. e.* Typhimurium, Santiviago et al. [23] were able to establish that stm3119 mutants lacked “fitness” in this experimental system. The presence of the three genes as part of SPI-13 in *S. e.* Typhimurium was demonstrated by Haneda et al. [24]. The authors also established that stm3117, 3118 and 3119 deletion mutants showed a low competitive index in a mouse infection model. Transposon-generated mutant pools were utilized by Chaudhuri et al. [25] to infect chickens, pigs, cattle and mice and to determine the relative fitness of the mutants. The findings were not completely in line with those from other authors in that insertions in the stm3117 open reading frame (ORF) did not impact fitness in the four animal types; however, insertions in the stm3118 equivalent caused attenuation. Attenuation of stm3119 insertion mutants was also observed in cattle and mice, but not in chickens and pigs. Finally, Elder et al. [26] found that SPI-13 deletion mutants of *S. e.* Enteritidis were impaired in their ability to proliferate in the ceca, liver and spleen of streptomycin-pretreated mice. These mutants also showed reduced intestinal inflammation. Survival in murine macrophages was also impaired. In contrast, the SPI-13 deletion mutants did not experience any effects on survival in avian macrophages or on survival in organs. 

Despite their relatively high similarity to the *ripABC*-encoded proteins of *Y. pestis*, genome annotations usually identify STM3117 and its equivalents in other *Salmonella* as a lactoglutathione lyase, STM3118 as an acetyl-CoA hydrolase/transferase and STM3119 as monoamine oxidase family dehydratase. Given this annotation, Chakraborty et al. [27] hypothesized that STM3117 was involved in the detoxification of methylglyoxal, a compound produced by some intestinal bacteria under anaerobic conditions [28]. The authors compared the growth of *S. e.* Typhimurium NCTC 12023 with that of an stm3117 deletion mutant in the presence of exogenous methylglyoxal and found an approximately 15% decrease in survival in the mutant in the presence of 5 μM methylglyoxal, after 3 h of exposure to the compound. Interestingly, the mutant also exhibited a somewhat slower growth and final optical density in the nutrient-rich media, LB and Terrific Broth, but not in minimal medium with glucose. The authors speculated that this inhibition is caused by the metabolism of threonine and glycine present in the rich media to produce aminoacetone, which is then transaminated to methylglyoxal by STM3119 (amine oxidase). The authors expressed STM3117 in *E. coli* and claimed that the protein was associated with the outer membrane. When the enzyme kinetics for the conversion of methylglyoxal glutathione hemiacetal *S*-D-lactoglutathione was measured, the authors found that activity was stimulated by Co^2+^ and obtained a K_m_ of 0.588 mM^−1^ (sic) and a V_max_ of 13.12 μmol min^−1^ per mg of protein. In comparison, the corresponding enzyme from *Y. pestis* exhibited a K_m_ of 2.2 mM and a V_max_ of 203.7 U per mg of protein for the conversion of itaconate to itaconyl-CoA. No stimulation by divalent ions was observed. As further evidence for the role of STM3117 enzyme in methylglyoxal detoxification, Chakraborty et al. [27] transferred a plasmid encoding STM3117 into *S. e.* Typhi, which does not encode an equivalent enzyme, and they obtained a strain whose growth was (somewhat) less sensitive to the presence of 10 μM methylglyoxal. At this time, it is difficult to reconcile the data obtained by Chakraborty et al. [27] with those suggesting itaconic acid degradation activity by the enzymes. 

Although unambiguous biochemical evidence for the enzymatic function of STM3117, 3118 and 3119 in itaconic acid degradation is not available, their amino acid sequence similarity with proven itaconic acid degradation enzymes from *Y. pestis* has led to the designation of the corresponding genes as *ict*, *ich* and *ccl* by Hammerer et al. [29]. These authors sought to inhibit itaconic acid degradation in *S. e.* Typhimurium ATCC 14028 and thus render the cells sensitive to itaconic acid by supplying the bacteria with synthetic compounds consisting of an itaconic acid-mimicking moiety linked to a pantothenate group. These compounds would act as prodrugs that would be converted to their CoA derivatives within the pathogen. In turn, these derivatives would inhibit itaconic acid degradation and render the cells sensitive to itaconic acid. One of the compounds was indeed able to confer itaconic acid sensitivity to the pathogen in vitro. 

## 3. Distribution of Itaconic Acid Degradation Genes within the Genus *Salmonella*

Santiviago et al. [23] and Haneda et al. [24] pointed out that stm3117–3119 were not present in the genomes of *S. e.* Typhi, and Espinoza et al. [30] included *S. e.* Paratyphi A, but these are not the only serovars that do not harbor these genes, as some unpublished work carried out in our laboratory demonstrates. When conducting growth studies in M9 minimal medium, with acetate as the carbon source, in the manner described by Hammerer et al. [29], we found that several poultry isolates were inhibited by itaconic acid concentrations in the range at which this acid could potentially be found in activated macrophages. These isolates belonged to serovar Kentucky, Senftenberg and Worthington. With one exception, strains belonging to serovar Typhimurium did not exhibit this growth inhibition. As can be seen in Figure 1B,C, *S. e.* Typhimurium DT104 was not inhibited by concentrations of itaconic acid of less than 12.5 mM in the presence of acetate. In the presence of pyruvate as a carbon source, no inhibition by itaconic acid concentrations less than approximately 40 mM was observed with this strain. In contrast, *S. e.* serovar Kentucky T-30, a poultry isolate, was inhibited by concentrations as low as approximately 2 mM. Similar to the observations of Hammerer et al. [29], some growth stimulation was observed with the Typhimurium strain in the presence of around 5 mM itaconic acid in the acetic acid-containing medium. The reason for this occurrence remains to be explained, as does the more extensive growth stimulation of this strain in the presence of 5 to 25 mM itaconic acid with pyruvate as carbon source. As shown in Figure 1A, itaconic acid at concentrations lower than 12.5 mM was not inhibitory to S. e. Kentucky T-30 when cells were grown in minimal medium with glucose as a carbon source. The apparent difference in growth between the two serovars at itaconic acid concentrations above 6.25 mM could possibly be explained by the lower pH of the medium, which could favor the Kentucky isolate, as previously observed [31]. Results from PCR analyses using primers specific for the three genes paralleled those obtained with the growth studies (our unpublished data) in that itaconate sensitivity was correlated with the absence of the three genes. 

A search by the author of this review of the 1063 complete *Salmonella* spp. genomes available in Genbank revealed that the three genes, or sequences similar to them, are not present in *S. bongori* or the *Salmonella* subspecies salamae and indica (Table 1). Of the 102 subspecies of enterica serovars represented in the genome collection, the majority, 75, harbor *ict*, *ich* and *ccl*. It is noteworthy that the sequenced isolates belonging to subspecies enterica serovars most commonly associated with human infections, such as serovars Enteritidis, Newport, Typhimurium, Javiana, 1,4,[5],12:i:-, Infantis, Muenchen, Montevideo, Braenderup and Thompson [32], almost all harbor the three genes. This gene arrangement is illustrated by the example of *S. e.* Typhimurium LT2 in Figure 2. In the case of serovars Typhimurium, Infantis, Mbandaka and Stanleyville, genomes without the three genes were also found. It is possible that these genomes came from isolates that were misidentified serologically, but it is also possible that some heterogeneity exists within certain serovars. 

In contrast, several serovars isolated from clinical or non-clinical animal sources in the greatest numbers, including serovars such as Kentucky, Senftenberg, Worthington, Agona, Albany and Ouakam [33], do not carry these genes or even partially matching sequences. In 25 of the serovars represented in the genome collection, the genomic regions where the itaconate degradation genes would be located harbor sequences that, as their most prominent feature, contain two putative bacteriocin immunity genes (Figure 2), as previously pointed out for *S. e.* Typhi CT18 [34]. The question arises as to whether the two groups of *Salmonella* serovars acquired their respective genes in two independent events or if one replaced the other at a later time. The fact that some isolates do not carry the itaconic acid degradation genes or the bacteriocin immunity genes indicates that these genes are not essential for survival. 

## 4. *Salmonella* and Bacteriocins

Bacteriocins, ribosomally synthesized antimicrobial polypeptides with a usually narrow inhibitory spectrum, are produced by many bacteria, including those frequently found in the intestinal tract (reviewed in [35]). While a number of publications, such as [36,37], claim to have identified bacteriocins or bacteriocin-like substances, produced by unrelated microorganisms such as lactobacilli, that can inhibit *Salmonella* in vitro, it is likely that most bacteriocins that are inhibitory to *Salmonella* sp. are produced by closely related species, such as colicin-producing *E. coli* or other *Salmonella* strains. In fact, by the 1950s, it was fully recognized that some *Salmonella* isolates produced colicin-like compounds [38] and that certain *Salmonella* isolates were sensitive to colicins [39]. A study of 972 *Salmonella* isolates belonging to several serovars found 101 isolates that showed colicin-type antimicrobial activities [40]. Similarly, tests on 1825 *Salmonella* isolates revealed 98 with colicin-type activities [41]. Studying members of serovar Typhimurium, Barker [42] found that among 4481 isolates, 531 produced colicin-like compounds. Bacteriocins from *Salmonella* were initially termed salmonellins [43], later, salcols [44], and more recently, salmocins [45]. In the latter publication, the authors showed that some salmocins exhibit activity against a broad range of *Salmonella* serovars. 

Bacteriocin production is frequently associated with Group I1 (IncI1) plasmids in *Salmonella enterica* [46,47], but, like in *E. coli* [48], other types of bacteriocin-encoding plasmids are likely also present in *Salmonella*. For example, some serovar Kentucky isolates carry a plasmid encoding colicin V [49,50]. Usually, the gene encoding the bacteriocin is located in a cluster that also contains an immunity gene, a lysis gene or genes facilitating export [51]. Some bacteriocins, such as Microcin H47 and its immunity and other associated genes, are chromosomally encoded [52].

Clustering of colicin and immunity genes makes sense as coordinate synthesis is crucial to avoid the killing of the cell by its own bacteriocin. In the case of the genomic region containing the two bacteriocin immunity genes, no evidence for bacteriocin-producing genes could be found. It is possible that such genes are located somewhere else in the genome, but it is also possible that the two immunity genes are retained in the genome without the genes that they are usually clustered with. It is currently not known if these immunity genes are actually producing functional proteins, but their highly conserved sequences among members of the enterica subspecies of *S. enterica* that carry them could suggest that they likely have a useful function. 

The region containing the bacteriocin immunity gene and the itaconic acid degradation gene region have G + C contents of 38.3% and 47.2%, respectively. Thus, their G+C contents are lower than that of the overall *Salmonella* genome, which has a G + C content of 52.2%, and these sequences could therefore have been acquired from other bacteria (reviewed in [53]). The two immunity genes share 51% identity. Based on results obtained with BlastP, the putative protein encoded by the immunity gene closest to tRNA^phe^ has 59.8% identity to chain A of the colicin-e7 immunity protein (Imme 7) from *E. coli*, and the second gene encodes a bacteriocin immunity protein targeting a colicin with DNase activity with 79.8% identity to its *E. coli* counterpart. 

Assuming that the immunity proteins are actually synthesized by the *Salmonella* strains that harbor the corresponding genes, it is likely that they serve the function of protecting these strains from attack by specific bacteriocins. It was already shown with the early studies on colicins [30] that some of the colicins were able to inhibit *Salmonella* in vitro. More recently, Wooley et al. [54,55] demonstrated the killing of *Salmonella* isolates by an inactivated culture of a microcin 24-producing *E. coli* strain. Zihler et al. [56] tested the in vitro activity of *E. coli* producing microcin 24 or microcin B17 against a collection of *Salmonella* isolated from clinical cases and representing 11 serovars and found that all or almost all of their isolates were inhibited by the two respective microcins. 

In vivo studies attempting to demonstrate the effect of bacteriocin-producing *E. coli* on *Salmonella* have also been carried out. Wooley et al. [54] administered a microcin 24-producing *E. coli* continuously to chickens infected with *S. e.* Typhimurium, and the authors were able to show a significant reduction in the pathogen in the intestinal tract under these conditions. Studies with bacteriocin-producing *Listeria monocytogenes* [57] and *Enterococcus faecalis* [58] have provided evidence that such strains can alter the intestinal microbiota in favor of the bacteriocin producers. Similarly, *Salmonella* can change the intestinal environment. In a mouse model, Stecher et al. [59] demonstrated that *S. e.* Typhimurium can cause the appearance of blooms of the pathogen and resident *E. coli* in the inflamed gut and that during such blooms, the conjugative transfer of colicin plasmid p2 from *S. e.* Typhimurium to *E. coli* occurs at high rates. During such enterobacterial “blooms”, when competition between *E. coli* and *S. e.* Typhimurium can be expected, bacteriocin production gives a competitive advantage to the producer, which was shown by Nedialkova et al. [60]. Similarly, Sassone-Corsi et al. [61] demonstrated that microcin-producing *E. coli* Nissle 1917 was able to reduce intestinal colonization by *S. e.* Typhimurium in the inflamed gut of mice. 

Thus, under conditions in which bacteriocin-producing *E. coli* are present in high numbers, it might be advantageous for *Salmonella* to have the means to prevent an attack by bacteriocins by expressing immunity genes. The production of immunity proteins might provide a sufficiently competitive edge, making it unnecessary to also produce bacteriocins and other proteins necessary for bacteriocin production and lysis or export. 

## 5. Why Itaconic Acid Degradation or Bacteriocin Immunity Genes?

Why some *Salmonella* serovars have acquired genes likely required for the degradation of itaconic acid while others have incorporated DNA that could encode bacteriocin immunity genes remains to be explained. We have previously proposed that poultry isolates of *S. e.* Kentucky show properties that could be interpreted as being advantageous when located in the cecum (e.g., acid response) and disadvantageous for host cell invasion (absence of a number of virulence genes) [31]. It is possible that serovar Kentucky and some other serovars spend more time in the lumen or the mucus layer of intestinal sites and have a greater need to defend themselves against bacteriocins produced by *E. coli* or other *Salmonella* residents. We and others have shown that serovar Kentucky is capable of invading human ileocecal adenocarcinoma cells (HCT-8 ATCC CCL-244) [31], chicken macrophage cells (HD11) [62] or human colon cells (Caco-2) [63], but those assays were terminated after, at most, 24 h, and longer-term intracellular survival was not tested. Possibly, lacking itaconic acid degradation genes would limit intracellular proliferation in comparison to that of serovars that harbor these genes. Alternatively, it is possible that strains without itaconate degradation genes have means to reduce or eliminate the threat posed by itaconic acid. For example, this may be the case if uptake of the acid was restricted (a possibility suggested by the data presented by Martin et al. [14]) or if isocitrate lyase was less sensitive to the presence of the acid. Simply “avoiding” hosts that produce itaconic acid or tissues and cells where it is found in inhibitory concentrations might be another strategy of coping with the acid. 

It is interesting that the animal host-adapted serovars Typhimurium, Gallinarum and Cholerasuis harbor the itaconic acid degradation genes, whereas the human host-adapted serovars Typhi and Paratyphi A do not. Whether or not this means that these two serovars are as sensitive to itaconic acid as, for example, serovar Kentucky, is not known. Perhaps these serovars do not spent a lot of time in or near macrophages and thus itaconic acid but rather quickly disseminate to other cell types that do not produce this acid. It is known that *S. e.* Typhi can persist for long periods in the gall bladder [64,65]. Even if a functioning glyoxylate shunt is required at this site, it is possible that the pathogen does not encounter itaconic acid there. In addition, perhaps macrophages from different hosts produce different amounts of itaconic acid, as was observed by Michelucci et al. [6], who measured itaconic acid concentrations of 8 mM and 60 μM in mouse and human macrophages, respectively. 

In a few of the analyzed genomes, neither itaconic acid degradation genes nor the bacteriocin immunity genes are present (Table 1, Figure 2). They are absent from the genomes of *S. bongori* and the *S. enterica* subspecies salamae and indica, as well as from *S. enterica* subspecies enterica serovars Adjame, Indiana, Manchester, Ohio and Sanjuan. Some serovars have members with the itaconic acid degradation or bacteriocin immunity genes but also a few without them—for example, serovars Derby, Kentucky, Macclesfield, Montevideo, Mbandaka, Rissen, Senftenberg and Typhimurium (Table 1). In what way the absence of these genes impacts the survival and proliferation of these isolates is difficult to say, but the fact that such strains could be isolated indicates that they found ecological niches to survive and proliferate in. 

## Figures and Tables

**Figure 1 genes-11-00797-f001:**
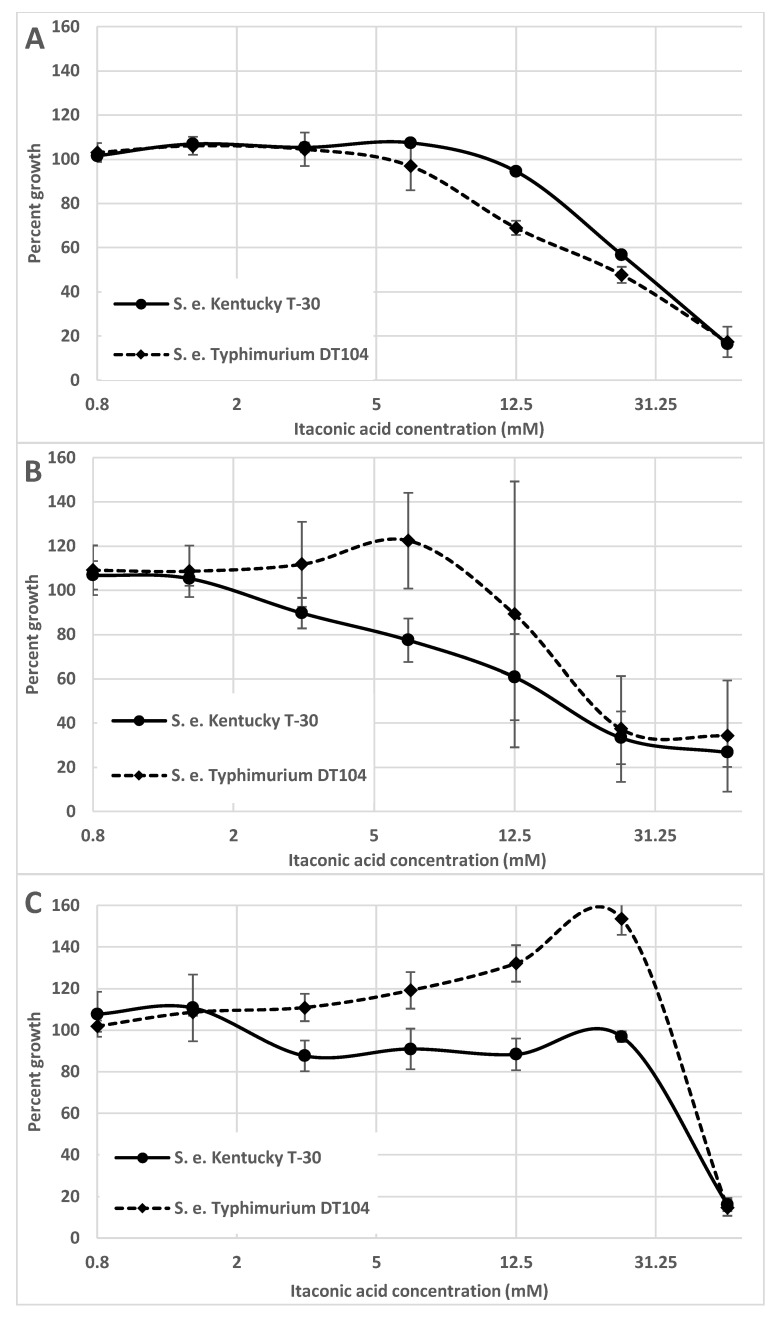
Percentage growth in M9 minimal medium containing various concentrations of itaconic acid, expressed as percentage of growth in the medium without itaconic acid. Cells were grown as described by Hammerer et al. [29]. (**A**) Growth percentages in M9 minimal medium containing 1% glucose. (**B**) Growth percentages in M9 minimal medium containing 0.4% acetate. (**C**) Growth percentages in M9 minimal medium containing 0.4% pyruvic acid. The pH of the M9 minimal media was adjusted to pH 6.8 prior to the addition of itaconic acid. Results are averages of six independent growth studies. The error bars indicate standard deviation. Student *t*-test analysis for the differences in means indicated *p*-values of 0.035, <0.001 and 0.079 for itaconic acid concentrations of 3.125, 6.25 and 12.5, respectively, for growth in acetate-containing medium and of <0.001 for itaconic acid for itaconic acid concentrations of 3.125, 6.25, 12.5 and 25 mM for cells grown in medium with pyruvate.

**Figure 2 genes-11-00797-f002:**
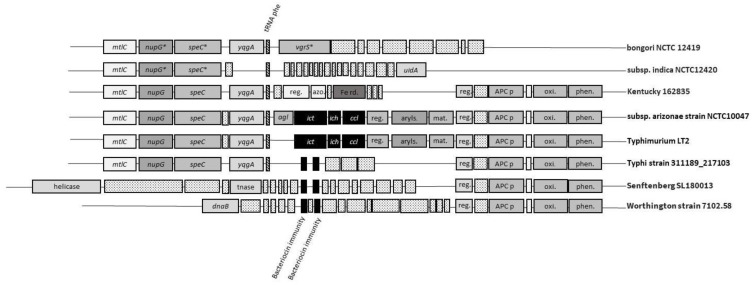
Examples of gene organization in the tRNA^Phe^ region of *Salmonella* spp. Genes are drawn approximately to scale; however, intergenic regions are not. *indicates potential frameshifts within the open reading frame (ORF). ORFs indicated by dotted boxes represent pseudogenes or ORFs without known functions. Gene names and ORF designations (encoded protein in parentheses): *agl*, (lactoglutathione lyase); APCp, (APC family permease); aryls., (arylsulfatase); azo, (FMN-dependent NADH-azoreductase); *ccl*, ((*S*)-citamalyl-CoA lyase); *dnaB*, (replicative DNA helicase); Fe red., (iron reductase); *ich*, (itaconyl-CoA hydratase); *ict*, (iconate CoA transferase); mat., (anaerobic sulfatase maturase); *mtlC*, (murein transglycosylase C); *nupG*, (nucleoside transporter); oxi, (FAD-binding oxidoreductase); phen., (NAD-dependent phenylacetaldehyde dehydrogenase); reg., (LuxR- or LysR-type regulatory proteins); speC, (ornithine decarboxylase); uidA, (β-glucuronidase); *vgrS*, (Type VI secretion system tip protein); *yqgA*, (putative transporter).

**Table 1 genes-11-00797-t001:** Number of complete genome sequences of Salmonella isolates belonging to different species, subspecies and serovars with genes for itaconate degradation (Column A), for bacteriocin immunity (Column B) or neither (Column C). A total of 1063 Salmonella genomes was searched using Blast N.

Species	Subspecies	Serovar	A *	B *	C *	Species	Subspecies	Serovar	A *	B *	C *
*S. bongori*			0	0	12	*S. enterica*	*enterica*	Hillingdon	0	1	0
*S. enterica*	*houtenae*		4	0	0			Hvittingfoss	2	0	0
	*diarizonae*		10	0	0			Indiana	0	0	6
	*arizonae*		8	0	0			Infantis	10	3	0
	*salamae*		0	0	11			Iverness	1	0	0
	*indica*		0	0	9			Java	1	0	0
	*enterica*	unkown	93	21	9			Javiana	2	0	0
		1,4,[5],12:i:-	29	0	0			Johannesburg	1	0	0
		4,[5],12:i:-	8	0	0			Kentucky	0	4	1
		Abaetetuba	1	0	0			Koessen	1	0	0
		Aberdeen	1	0	0			Krefeld	0	1	0
		Abony	1	0	0			Macclesfield	1	0	1
		Adjame	0	0	11			Manchester	0	0	1
		Agona	0	14	0			Manhattan	1	0	0
		Albany	0	2	0			Mbandaka	1	0	2
		Anatum	29	0	0			Mikawasima	2	0	0
		Antsalova	1	0	0			Milwaukee	0	1	0
		Apapa	0	1	0			Minnesota	2	0	0
		Bardo	1	0	0			Moscow	1	0	0
		Bareilly	29	0	0			Muenchen	3	0	0
		Bergen	0	1	0			Muenster	3	0	0
		Berta	1	0	0			Montevideo	18	0	1
		Birkenhead	1	0	0			Newport	32	0	0
		Blegdam	1	0	0			Ohio	0	0	1
		Blockly	1	0	0			Onderstepoort	1	0	0
		Borreze	0	1	0			Oranienburg	2	0	0
		Bovismorbificans	1	0	0			Ouakam	0	0	1
		Brancaster	0	1	0			Panama	1	0	0
		Brandenburg	3	0	0			Pomona	2	0	0
		Braenderup	3	0	0			Poona	2	0	0
		Bredeney	3	0	0			Paratyphi A	0	6	0
		California	1	0	0			Paratyphi B	1	0	0
		Carmel	1	0	0			Paratyphi C	1	0	0
		Cerro	2	0	0			Pullorum	4	0	0
		Chester	1	0	0			Quebec	1	0	0
		Cholerasuis	4	0	0			Rissen	0	1	3
		Concord	1	0	0			Rough C:-:-	2	0	0
		Corvallis	0	2	0			Saintpaul	8	1	0
		Crossness	1	0	0			Sanjuan	0	0	1
		Cubana	0	1	0			Senftenberg	0	14	1
		Daytona	1	0	0			Schwarzengrund	3	0	0
		Derby	0	2	3			Sloterdijk	1	0	0
		Djakarta	1	0	0			Stanley	2	0	0
		Dublin	12	0	0			Stanleyville	4	1	0
		Enteritidis	241	0	0			Sundsvall	1	0	0
		Florida	1	0	0			Tennessee	0	7	0
		Fresno	0	1	0			Thompson	9	0	0
		Gallinarum	5	0	0			Typhi	0	122	0
		Gaminara	2	0	0			Typhimurium	96	1	1
		Give	2	0	0			Virchow	2	0	0
		Goaldcoast	3	0	0			Wandsworth	1	0	0
		Hadar	1	0	0			Weltevreden	5	0	0
		Havana	0	1	0			Worthington	0	2	0
		Hayindogo	1	0	0			Yovokome	1	0	0
		Heidelberg	33	0	0						

* **A**, possession of *ict*, *ich* and *ccl;*
**B**, possession of bacteriocin immunity genes; **C**, other genes. The data were generated by downloading *Salmonella* genomes from Genbank and removing genomes that were not designated as complete. The remaining chromosomal sequences were searched with the *ict*, *ich* and *ccl* sequences, as well as the sequences of the putative bacteriocin immunity genes. Blast N results were either matches with >98% identity or no matches (“no significant similarity found”).

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
