# Peer review of "Salmonella enterica’s “Choice”: Itaconic Acid Degradation or Bacteriocin Immunity Genes"

_genes, 2020, doi:10.3390/genes11070797_

Round 1
Reviewer 1 Report
The manuscript has been submitted as a review article. In addition to conducting a typical review, however, the Author reveals and discusses new results of cultivations experiments and bioinformatics analyses. The title of the manuscript highlights the hypothesis formulated by the Author on the basis of literature survey and the newly generated datasets. I have doubts whether the article can still be categorized as a "review", but at the same time I feel the presented results would not be sufficient to provide the basis for preparing an original research article. Having said that, even if the approach to review the topic is far from being a typical one, I still find it interesting. On the other hand, since new experiments were in fact carried out, the Materials and methods section would be recommended to specify the cultivation procedures and the details of bioinformatic analyses. Including the Materials and methods in the Supplementary materials may be considered.
The manuscript fragments referring to previously published results (i.e. the true “review”) are well written and clear. The parts regarding novel results, especially the cultivation study, are the ones that need to be clarified before the manuscript can be considered for publication.
In my opinion, the part regarding growth experiments (fig. 1) requires thorough revision (lines 142-148). First of all, this fragment should be somehow separated from the remaining text to inform the reader that the new data is shown. There ought to be a clear boundary between the previously reported research and the original contribution. In the current version, the text reads “These isolates belonged to serovar Kentucky, Senftenberg and Worthington.” (line 141) and then, suddenly, the S. e. Typhimurium DT104 appears with respect to which the data presented in fig. 1 was gathered. This may seem a bit chaotic and confusing at first (a short introduction to perfomed experimental work would not be a bad idea at this point). The next important issue concerns the discussion and interpretation of results. What is meant by “low concentration of itaconic acid” (line 143)? Which data points in fig. 1 are referred to here? It is stated that “S. e. Typhimurium DT104 was not inhibited by low concentrations of itaconic acid in the presence of acetate or pyruvate as carbon source, whereas S. e. serovar Kentucky T-30, a poultry isolate, was inhibited.” In my opinion, these observations need to be supported by statistical analysis. Where the results obtained in the presence of itaconic acid significantly different from the ones recorded without it? Providing the p-values would make the data much easier to interpret. Importantly, the error bars in fig. 1 are not described. Was the experiment performed in triplicates? How is “growth” measured in the study? Figure caption needs to be updated to include all the required information. Shortly speaking, stating that one variant was inhibited, while the second one was not, would require statistical justification. Furthermore, whenever the “Percent growth” exceeds 100% (at times it reaches as much as 160%), a comment should be included on why such behavior could be observed.
Table 1 gathers the data regarding the presence or absence of certain genes in Salmonella genomes. It is also claimed that (line 159): “A search of 1,063 complete Salmonella spp. genomes available in Genbank revealed that the three genes are not present in S. bongori and the Salmonella subspecies salamae and indica (Table 1).” In my opinion, the criteria according to which the gene was regarded as absent or present should be provided in the Materials and methods section or at least in the table footnote (e.g. E value, % identity). In other words, the cutoff values defining the actual BLAST hits ought to be specified to interpret the results correctly.
Even though the hypothesis formulated by the Author has not been proved in any way, it is still a valuable and interesting perspective on the topic addressed in the submitted manuscript.
Author Response
I would like to thank the reviewer for her/his constructive criticism of the manuscript. Although the inclusion of the data from my laboratory was primarily intended to supplement information in the literature, I agree that these data need to be more clearly labeled as new data and that more information on the methodology needs to be provided. Therefore, the following changes were made to the manuscript:
- L. 140 - 167: Changes were made to more clearly indicate that this section concerns data generated by the author. Also, a reference for the methodology was included and a sentence that leads to the example of S. e. Typhimurium DT104 in Fig. 1 was added. The qualitative statements regarding the itaconic acid concentrations were replaced with quantitative data.
- L. 169 - 178: The legend for Fig. 1 was expanded to include a reference for the methodology and information on statistical analyses.
- L. 170 - 198: Modifications were made to better indicate that this section contains data from the author. Also, it is now indicated that the sequence comparisons showed either complete matches or no matches at all. (No partial identities.)
- L. 203 - 207: The footnote to Table 1 was expanded to include details on the methodology used for the sequence comparisons and level of sequence matches observed.
Reviewer 2 Report
This mini review gives a detailed overview of the recent literature about the presence of (or lack of) genes for itaconic acid degradation and/or bacteriocin immuniy in bacteria Salmonella enterica. The current known literature is well presented and commented. Author gives his insight in the subject and some of his results are presented as well. The only thing the manuscript lacks is a better structure. The journal has no strict demands on the form between the front and the back matter but dividing the text into few sections would make it more comprehensive and easier to follow.
Line 34. Please add full description before abbreviation i.e. "lipopolysaccharide (LPS)-trated..."
Author Response
I appreciate the concern regarding the structure of the manuscript. In response, I have introduced titles to individual sections of the manuscript in the hope that in this way, the information is easier to follow.
L. 34: "lipopolysaccharide" was added in front of the abbreviation.